# Minimally Invasive Hypoglossal Nerve Stimulator Enabled by ECG Sensor and WPT to Manage Obstructive Sleep Apnea

**DOI:** 10.3390/s23218882

**Published:** 2023-11-01

**Authors:** Fen Xia, Hanrui Li, Yixi Li, Xing Liu, Yankun Xu, Chaoming Fang, Qiming Hou, Siyu Lin, Zhao Zhang, Jie Yang, Mohamad Sawan

**Affiliations:** 1Zhejiang University, Hangzhou 310024, China; xiafen@westlake.edu.cn; 2CenBRAIN Laboratory, School of Engineering, Westlake University, Hangzhou 310024, China; hanrui.li@kaust.edu.sa (H.L.);; 3SAMA Labs, Computer, Electrical and Mathematical Science and Engineering (CEMSE) Division, Department of Electrical and Computer Engineering, King Abdullah University of Science and Technology (KAUST), Thuwal 23955-6900, Saudi Arabia; 4State Key Laboratory of Superlattices, Microstructures Institute of Semiconductors, Chinese Academy of Sciences, Beijing 100045, China; liyixi22@mails.ucas.ac.cn

**Keywords:** OSA, HGNS, ECG, CNN, WPT, rectifier, power management module, BPSK, stimulator, implant

## Abstract

A hypoglossal nerve stimulator (HGNS) is an invasive device that is used to treat obstructive sleep apnea (OSA) through electrical stimulation. The conventional implantable HGNS device consists of a stimuli generator, a breathing sensor, and electrodes connected to the hypoglossal nerve via leads. However, this implant is bulky and causes significant trauma. In this paper, we propose a minimally invasive HGNS based on an electrocardiogram (ECG) sensor and wireless power transfer (WPT), consisting of a wearable breathing monitor and an implantable stimulator. The breathing external monitor utilizes an ECG sensor to identify abnormal breathing patterns associated with OSA with 88.68% accuracy, achieved through the utilization of a convolutional neural network (CNN) algorithm. With a skin thickness of 5 mm and a receiving coil diameter of 9 mm, the power conversion efficiency was measured as 31.8%. The implantable device, on the other hand, is composed of a front-end CMOS power management module (PMM), a binary-phase-shift-keying (BPSK)-based data demodulator, and a bipolar biphasic current stimuli generator. The PMM, with a silicon area of 0.06 mm${}^{2}$ (excluding PADs), demonstrated a power conversion efficiency of 77.5% when operating at a receiving frequency of 2 MHz. Furthermore, it offers three-voltage options (1.2 V, 1.8 V, and 3.1 V). Within the data receiver component, a low-power BPSK demodulator was ingeniously incorporated, consuming only 42 μW when supplied with a voltage of 0.7 V. The performance was achieved through the implementation of the self-biased phase-locked-loop (PLL) technique. The stimuli generator delivers biphasic constant currents, providing a 5 bit programmable range spanning from 0 to 2.4 mA. The functionality of the proposed ECG- and WPT-based HGNS was validated, representing a highly promising solution for the effective management of OSA, all while minimizing the trauma and space requirements.

## 1. Introduction

Obstructive sleep apnea (OSA) is a sleep disorder characterized by the partial or complete blockage of the upper airway. The most-common symptoms are loud and frequent snoring, pauses in breathing during sleep, dry mouth, excessive daytime sleepiness, and morning headaches. This widespread condition places a significant burden on the healthcare system and affects nearly one-billion people globally [1]. If left untreated, it can lead to serious health or quality-of-life-related problems, such as strokes, hypertension, cardiovascular disease, depression or anxiety, and hypnosis. Treatment options for OSA can be broadly categorized into non-invasive and invasive methods [2]. The most-widely used non-invasive method is continuous positive airway pressure (CPAP) [3]. However, wearing a bulky CPAP device during treatment can be uncomfortable for some patients.

The advancement in neuroscience and electronics has led to the increasing exploration of hypoglossal nerve stimulators (HGNSs) as a potential treatment option for OSA [4]. When a patient is experiencing OSA, the hypoglossal nerve can be electrically stimulated to contract the genioglossus muscle, thereby opening the blocked upper airway. It provides a success rate of approximately 76%, which is highly comparable with CPAP [5].

A conventional HGNS consist of three components: a breathing sensor, a generator, and stimulating electrodes, as is seen in Figure 1a [6]. Each of these three parts is located in a different area of the body and is connected through leads. Using the breathing sensor, the generator analyzes whether the patient is under OSA or not in order to determine whether to send a mild stimulation command to the stimulation lead. As a result of this stimulation, the upper airway can remain open during sleep. The three main parts require at least three implantation surgeries, increasing the risk of safety issues [6]. For control and communication purposes, an external remote controller with Redtooth functionality is employed to trigger and adjust this implantable stimulator. It is important to note that accommodating both the battery and breathing sensor within the implantable device consumes a substantial amount of invasive space. Furthermore, the battery in this system has a limited lifespan and may require a second surgery for replacement, resulting in additional trauma for the patient [7].

Decades of research on implantable electrical stimulation have brought significant benefits to patients with neural or muscular disorders. Examples of mature implantable biomedical devices include cardiac pacemakers, while other areas of active research include visual prostheses and cochlear implants [8]. These implants utilize the wireless power transfer (WPT) technique to divide the device into parts internal and external to the body to minimize the volume of the implantable components. However, the research on HGNSs remains largely focused on treatment outcome analysis, the subject group, surgery procedure improvement, and comparison with other techniques [7]. The improvement of the HGNS system in terms of structure is rarely mentioned.

In this context, we introduce an innovative ECG- and WPT-based hypoglossal nerve stimulator, referred to as the EWHGNS, with the aim of overcoming the limitations inherent in conventional HGNS systems. The proposed EWHGNS device is designed to be both compact and highly flexible for communication and stimulation purposes. As shown in Figure 1b, the EWHGNS comprises two core components: an external controller and an implantable generator. The external controller collects the ECG data and employs a CNN algorithm for OSA detection. Upon identifying an apnea episode, this user interface controller promptly transmits stimulating details and energy to the implant. To power the implantable generator, an inductive link is employed. This implantable generator is a highly integrated system-on-chip (SoC), which incorporates a power-management module to convert AC to DC. The generator is strategically connected to electrodes in close proximity to the hypoglossal nerve, minimizing the length of the leads and reducing the trauma area associated with implantation.

The EWHGNS system presents two noteworthy contributions. Firstly, our study presents a pioneering method in the treatment of OSA by introducing an ECG-based wireless-powered implantable hypoglossal nerve stimulator. This approach represents a pioneering advancement in the treatment of OSA. The EWHGNS system innovatively separates the conventional HGNS into internal and external components, effectively reducing the size of the implant. The external component of the system employs an ECG monitor and a CNN algorithm to analyze abnormal breathing patterns. This external device encompasses an OSA detector, an OSA digital signal processor (DSP), a BPSK modulator, and a power amplifier. The internal component of the EWHGNS system is responsible for receiving AC power and stimulation parameters from the external controller. It then converts the AC power into DC to provide power to the other circuit blocks housed within the EWHGNS. Meanwhile, the EWHGNS system performs demodulation of the stimulation parameters and executes biphasic constant current stimulation, contributing to its effectiveness in addressing OSA.

Secondly, each component within the implant showcases exceptional performance when compared to other research studies in their respective independent research fields. To illustrate this, the proposed ECG-based OSA detector attains an impressive accuracy of 88.68%, surpassing the k-nearest neighbor (k-NN), support vector machine (SVM), and recurrent neural network (RNN) classification methods in monitoring abnormal breathing patterns associated with OSA. Regarding the wireless power link, it achieved a measured power conversion efficiency (PCE) of 31.8% when tested with a 5 mm skin thickness and a receiving coil diameter of 9 mm. Notably, the power management module, occupying a compact area of only 0.06 mm${}^{2}$, delivers a remarkable power conversion efficiency of 77.5%. It also provides the flexibility to supply three distinct voltages (1.2 V, 1.8 V, 3.1 V). Lastly, a phase-locked-loop (PLL)-based BPSK demodulator was designed with an exceptionally low power consumption of only 42 μW.

The paper is organized into six sections. We describe in Section 2 the functional implementation requirements, which include the OSA detector, the hypoglossal nerve–electrode interface, the stimulation parameters, the voltage considerations, the WPT, the data exchange, and the architecture of the proposed EWHGNS system for OSA treatment application. Section 3 focuses on the methods and materials employed in the EWHGNS system, providing a detailed explanation of the design of each individual block. In Section 4, the paper presents the simulation and measurement results obtained for the EWHGNS system. Finally, we draw a conclusion in Section 5.

## 2. Considerations of the EWHGNS

In implantable biomedical devices, various essential functional blocks are commonly utilized, including the power management, data telemetry, control unit, and stimulator, to meet the requirements of low power consumption, a very small size, high reliability, and stability [9,10]. Nonetheless, it is crucial to recognize that there are specific challenges to be addressed depending on the unique requirements of each application. Variability can arise in aspects such as the interface with an external controller, the number of channels, the stimulation parameters, and the circuit design. As a pioneer in the quest to minimize the size of the hypoglossal nerve stimulator implant, we intend to provide detailed specifications related to the management of OSA from an application-oriented perspective in the forthcoming subsections.

### 2.1. OSA Detector

Detecting abnormalities in respiratory patterns accurately is crucial for assessing the efficacy of stimulation within the proposed EWHGNS. To begin, a thorough examination of wearable techniques suitable for monitoring OSA was conducted, as depicted in Table 1. Subsequently, Section 4 will provide a detailed illustration of the chosen method’s implementation.

Polysomnography (PSG) is considered to be the gold standard for diagnosing OSA [11]. It can simultaneously monitor multiple physiological modalities through various signal-acquisition devices during sleep. These modalities include brain activity (measured by electroencephalography or EEG), eye movements (measured by electrooculography or EOG), muscle activity (measured by electromyography or EMG), heart rate, breathing, oxygen levels, and snoring sounds [16]. However, obtaining a clinical diagnosis can be costly and time-consuming and typically requires laboratory testing, leading to challenges in monitoring patients in a residential setting with flexibility.

In accordance with previous research [12,13], it has been established that the electrocardiogram (ECG) stands out as the most-promising wearable approach for detecting obstructive sleep apnea (OSA), thanks to advancements in algorithms. The movement of the throat during each breathing cycle, as air fills and empties from the lungs, induces variations in the relative position of the electrodes concerning the heart. These variations can be effectively detected by monitoring changes in the amplitude information of the ECG [15]. The analysis of ECG data associated with OSA often involves the classification of changes in the amplitude of R- or S-waves, as well as the size of the QRS complexes, through ECG-derived-respiratory (EDR) analysis [17]. It is noteworthy that ECG-based approaches consistently demonstrate superior sensitivity and specificity in comparison to sound-based techniques and remote-based methods [13]. Recently, the diagnostic performance of oximetry for OSA has been enhanced through the implementation of a transformer neural network [14].

Sound-based respiratory monitoring, for instance, is susceptible to interference from a multitude of factors including coughing, wheezing, and ambient noises originating from both the surroundings and the body. Moreover, the scarcity of suitable acoustic datasets and the comparatively lower accuracy in comparison to the ECG-based method present a significant challenge when considering this method for our specific application [15]. Remote-based techniques, although promising for comfortable breathing monitoring, often exhibit limitations in accuracy due to the subject’s movements and activities [18]. Meanwhile, both sound-sensing and remote-based methods are in the research and development (R&D) stage, which cannot be applicable commercially in a short time [12]. Despite ongoing research on the reconstruction of the heart rate from photoplethysmography (PPG) while accounting for motion artifacts through a machine-learning-based approach [19], the diagnosis of OSA necessitates the development of a distinct algorithm for the transformation of the heart rate into the OSA diagnostic indicator. Furthermore, the accuracy of such an algorithm is not guaranteed.

Given these considerations, the ECG emerges as the optimal choice for the external OSA detector in our context.

### 2.2. Hypoglossal Nerve–Electrode Interface

In neuroprosthetic applications, the impedance of the electrode–nerve interface is critical to be considered as it directly influences the voltage supplied by the stimulator during maximum-intensity stimulation. This impedance can be simplified by employing an assembly of passive components that comprises a series-connected resistor ($R_{s}$) followed by a parallel combination of the resistor ($R_{p}$) and the capacitor ($C_{p}$) [10]. $R_{s}$ is the nerve impedance; $C_{p}$ is the double-layer nonlinear interface capacitance; $R_{p}$, representing a substantial impedance, mimics the hindrance encountered during charge transfer, which models the reduction and oxidation reactions within a state of electrochemical equilibrium. Then, this model is further simplified as a series resistor $R_{tot}$ and a series capacitor $C_{tot}$. The impedance $R_{tot}$ can vary within a broad range depending on the application, typically from 200 Ω to 10 kΩ [20]. The impedance of the hypoglossal nerve–electrode interface (HNEI) is estimated to have an $R_{tot}$ value of 0.8 kΩ, as reported in [21].

To further assess the impedance of the HNEI value, we measured the interface of an adult rabbit’s sciatic nerve and cuff electrodes. Figure 2 illustrates the experimental setup in animal testing using a rabbit model. The diameter of the sciatic nerve of this rabbit is 0.6 mm. The cuff electrode employed is crafted from high-purity platinum with a wire diameter of 0.1 mm and an electrode diameter of 1 mm. The spacing between two electrodes is maintained at 1 mm. Utilizing the VMP-300 measurement system, we determined the equivalent impedance values as follows: $R_{s}$ of 1.7 kΩ, $R_{p}$ of 8.4 MΩ, and $C_{p}$ of 7.8 μF. The measured impedance waveforms between two electrodes and the equivalent passive component circuit parameters are shown in Figure 3. Given that the diameter of the hypoglossal nerve in humans is around 4 mm, we opted for an $R_{tot}$ of 1 kΩ in this particular study. Considering 7.8 μF has almost a neglectable impact during stimulation as the voltage between the large capacitor is almost zero during one stimulation cycle; here, we selected a $C_{tot}$ of 400 nF.

### 2.3. Stimulation Parameters

Although specific optimal stimulation parameters for managing OSA have not been explicitly documented, valuable insights can be gained from previous research on hypoglossal nerve stimulation [22]. Based on this, we adopted a bipolar biphasic current generator in our experimental setup. The selected stimulation frequency was configured at 1.67 kHz (150 μs), comprising the anode stimulation (50 μs), discharge function (10 μs), cathode stimulation (50 μs), and discharge function (40 μs) per cycle. To regulate the amplitude of the stimulation intensity, we employed a digital-to-analog-converter (DAC)-based current generator. This device allows for programmable current intensities within a range of 0 to 2.4 mA with a resolution of 5 bit [23].

### 2.4. Voltage Consideration

The voltage between two electrodes under current stimulation can be expressed as:(1)$$V=I\times R_{tot}+\frac{I\times \Delta t_{pos}}{C_{tot}}$$
where *I* is the current. It is worth noting that the voltage value rises in direct proportion to the increase in current. To compute the minimum DC power requirement for the stimulator, it is imperative to set the maximum current at 2.4 mA. Given an impedance of $R_{tot}$ at 1 kΩ and a total capacitance $C_{tot}$ of 400 nF during the positive stimulation period ($\Delta t_{pos}$) of 50 μs, a minimum DC voltage of 2.7 V can be attained.

Within the EWHGNS system, the control unit typically operates with a power supply voltage of 1.2 V to minimize power consumption and align with technological process requirements. However, certain aspects of the system necessitate a higher voltage of 1.8 V. In the current reference circuit of a DAC-based current generator, a cascade current mirror structure is employed. This specific configuration serves to mitigate the channel length modulation effect and enhance the output impedance, ultimately leading to improved precision in the current reference generation. To support the high-precision current reference of this cascade structure, a voltage of 1.8 V is mandated. To accommodate these distinct voltage requirements, the EWHGNS system incorporates a dedicated power management module, which facilitates the provision of 1.2 V, 1.8 V, and 3.1 V supplies as needed for different components within the system [24].

### 2.5. Wireless Power Transfer

Inductive coupling is a prevailing WPT method used in implantable applications. The main goal of designing an inductive link is to minimize the size of the implantable coil while maximizing power transfer efficiency (PTE). PTE, a metric for WPT efficiency, is influenced by factors such as the dimensions of the coil and the depth of implantation.

Compensated capacitors have also been leveraged to boost PTE by utilizing resonance effects and mitigating system sensitivity to misalignment. The serial–parallel (SP) topology is generally preferred for the implantable part to supply high current compared with the serial–serial (SS) topology [25].

#### 2.5.1. Specific Absorption Rate

The specific absorption rate (SAR) quantifies the amount of wireless energy absorbed by the human body and should be carefully evaluated in the design of WPT systems for biomedical applications. It can be calculated using the equation:(2)$$SAR=\frac{\sigma \times E^{2}}{m_{d}}$$
where σ represents the tissue conductivity, *E* denotes the electric field, and $m_{d}$ represents the mass density.

#### 2.5.2. Operating Frequency

The direct correlation between tissue conductivity and operating frequency suggests that higher frequencies may lead to increased tissue conductivity, resulting in a potentially higher SAR [26]. To comply with SAR limitations, it is preferable to use lower operating frequencies. Improper and frequent switching of the active comparator in the PMM can result in multiple pulsing issues, which lead to the erroneous turn-on of large-sized power transistors and large leakage currents [27]. To address these issues, one potential approach is to lower the switching frequency.

Furthermore, higher frequencies are more sensitive to the △I/△t transient in rectifier design. When a comparator compares a sine wave voltage with the ground using a high-frequency switch, a significant resonance can occur because of rapid changes in the current and parasitic inductance [28]. To attain the optimal performance of the comparator, it is essential to mitigate the impact of parasitic inductance within the chip. This can be accomplished by incorporating additional parallel PADs and increasing the width of the layout line. However, it is important to note that implementing more parallel PADs and wider layout lines can result in a substantial increase in chip area. Lowering the operating frequency is an alternative method.

We employed a frequency of 2 MHz for both the power and communication bands in our proposed EWHGNS device to facilitate initial functional verification. Overall, this operating frequency has been carefully considered regarding the limitations of SAR, power management performance, and chip area requirements that were previously mentioned.

### 2.6. Data Exchange

The BPSK signal has been widely adopted as the downlink data communication through an inductive link for WPT [29]. In BPSK, binary data are encoded using two distinct phase states of the carrier signal. BPSK exhibits a constant envelope, ensuring that the power level of the transmitted signal remains consistent, thereby minimizing dynamic power loss [30]. In this context, a BPSK demodulator was selected to receive information from the external controller. For the EWHGNS system, the implant utilizes a 10 bit data format for unidirectional wireless communication, enabling constant current stimulation to ensure preliminary functional verification. Specifically, it includes 3 bit dedicated to determining the onset and offset of stimulation, an additional 5 bit for controlling the stimulation amplitude ranging from 0 to 2.4 mA, 1 bit for specifying the stimulation polarity (anode or cathode), and one bit to enable or disable charge balancing.

### 2.7. Architecture of the EWHGNS

To ensure the fulfillment of the aforementioned requirements and considerations, the main parameters are compiled in Table 2, and the global architecture is given in Figure 4.

The EWHGNS device comprises both an external controller and an implant. The external controller consists of several components, including an ECG-based obstructive sleep apnea (OSA) detector, an external control unit (CU), a modulator, and a power amplifier. The signal-processing workflow for ECG-based OSA detection encompasses the following steps in sequence: raw ECG signal acquisition, signal preprocessing, signal segmentation, feature extraction, and classification. The OSA detector provides real-time monitoring for patients, offering immediate feedback on the presence or absence of OSA (healthy). In instances where abnormal breathing patterns are detected, the OSA detector communicates signals to the CU. Subsequently, the CU initiates the transmission of power and stimulation parameters to the implant via a modulator and a power amplifier.

The implant is composed of four essential components: a power management module (PMM), a data exchange unit, an internal control unit (CU), and a biphasic current stimulating output stage module. This implant receives an alternating current (AC) supply through an inductive link, which is then transformed into a direct current (DC) voltage via a rectifier within the PMM.

However, this DC voltage is susceptible to fluctuations in the external controller’s power supply, as well as variations in the power conversion efficiency (PCE) of the WPT. To mitigate this, a bandgap reference (BGR) was engineered to generate a consistent amplitude, regardless of power supply variations. Following this, low-dropout (LDO) voltage regulators were designed to provide three distinct voltage levels (1.2 V, 1.8 V, and 3.1 V), requiring a high power supply rejection ratio (PSRR) to ensure a constant wireless voltage supply to the implants.

Simultaneously, the stimulation signal is demodulated by the data exchange unit, which subsequently transmits these data to the internal CU. The CU then forwards the data to the biphasic current stimulating output stage to execute the corresponding stimulating parameters for the hypoglossal nerve–electrode interface.

## 3. Design and Implementation of The Proposed System

### 3.1. OSA Detector

Figure 5 showcases the concrete implementation of the architecture of the ECG-based OSA detector.

The process commences with raw ECG recordings, which undergo signal preprocessing steps. These steps involve passing the signals through a low-noise amplifier and a low-pass filter to eliminate unwanted noise. Subsequently, an analog–digital converter (ADC) converts the preprocessed analog signals into digital format. Once the ECG signals are preprocessed, they undergo segmentation to extract relevant features.

The CNN divides the signals into segments, enabling the extraction of essential features. These features are then utilized in the classification of OSA behavior (normal or OSA). The processing element (PE) array is employed to execute the operations of the neural network. Each PE houses a local memory, which serves the function of storing the weight parameters. The matrix multiplication and accumulation are performed by this edge accelerator as the hardware implementation of the network. Regarding the network structure, five convolutional blocks are used in the ECG-based OSA detection task, which contains the convolutional, activation, and max pooling layer, respectively. Each convolutional kernel employs 1D-CNN, which is effective in capturing features from subsequences for one-dimensional temporal signals. The global average pooling layer and fully connected layer are performed on the previous feature map extracted by convolutional blocks. The output layer contains two neurons, which correspond to two classes (healthy, OSA) in the OSA detection task. The proposed method, as detailed in Section 4.1, achieved an accuracy of 88.68% on an open-source Apnea-ECG dataset, showcasing its effectiveness through cross-comparison with other approaches.

### 3.2. Wireless Power Transfer

We designed an inductive link with the aim of improving the PTE while taking into account the SAR considerations. The Federal Communications Commission (FCC) has established specific limits for the SAR of WPT in biological tissues. These limits are set at 1.6 and 2 W/kg for 1 and 10 g of tissue, respectively [31]. Our initial design involved the size of the coils and the depth at which the coil would be implanted. Since the target for hypoglossal nerve stimulation is located near the mandible where the skin depth is relatively shallow, we set the depth of the implantable coil at 5 mm. Additionally, the outer diameter of the receiving coil was initially established at 9 mm.

The PTE is based on the equivalent load resistance seen from the power supply [32], expressed as follows:(3)$$\eta_{{}_{\mathrm{WPT}}}=\frac{k^{2}Q_{1}Q_{L}}{1+k^{2}Q_{1}Q_{L}}\times \frac{Q_{2}}{Q_{2}+Q_{L}}$$
where the parameters are described in Table 3.

The expression for the PTE is obtained under resonant conditions by incorporating compensated capacitors. Notably, $\eta_{WPT}$ is dependent on the values of the inductances $L_{1}$ and $L_{2}$, mutual inductance *M*, the load resistance, and the operating frequency.

To precisely analyze the magnetic parameters of both coils, we used the COMSOL software (version 5.4) to construct a physical magnetic model. The optimal coil sizes were selected using the optimum PTE process [26], by tuning the coil radius and the number of turns.

### 3.3. Power Management Module

This power management module comprises an active doubler-based rectifier, a BGR, and three independent LDO regulators to provide the required regulated DC constant voltages, as reported in [24].

#### 3.3.1. Rectifier Circuit

The full-wave gate cross-coupled rectifier was preferred in this project due to its higher voltage conversion ratio when compared to a passive rectifier. Large-sized transistors were employed to minimize the threshold voltage of the main path transistors and reduce power conduction loss [25].

The proposed structure of the active doubler-based rectifier is shown in Figure 6. It includes two comparators (CMP1 and CMP2), large-sized NMOS M1 and PMOS M3, and auxiliary PMOSs M5 and M6 to ensure that the M3 current flows from the source to the drain. Additionally, it includes two transistors (M2 and M4) to start up the circuit, an off-chip bootstrap capacitor C3, and an off-chip filter capacitor C4. IGND is the abbreviation of implant-level ground, which differs from the external ground. Both comparators are powered by the output of the rectifier, denoted as $V_{REC}$.

The operation principle is described in several steps. First, when $V_{IN}$ is in the negative phase, which is less than $V_{DS}$ of M1, M1 powers up to charge C3. When $V_{IN}$ is at the lowest point, $V_{C3}$ equals $V_{IN,max}-V_{DS1}$ [33]. In Figure 6b, when $V_{VD}$ is less than the IGND, the current flowing through M10 is greater than that of the one flowing through the M7 and M9 branches. This results in a low-voltage level at the drain of M10 and a high-voltage output at $V_{CN}$ to power up M1. Subsequently, $V_{IN}$ enters a positive phase, and $V_{VD}$ is higher than zero volts. The output of CMP1 $V_{CN}$ is a low-voltage signal to turn off M1. With an increase in $V_{IN}$, $V_{VD}$ also increases to stabilize the charge of C3. If $V_{VD}$ is higher than the sum of the $V_{OUT}$ and $V_{SD}$ of M3, then M3 powers up to enable the current to flow to the loads $R_{L}$ and $C_{4}$. A schematic of CMP2 is presented in Figure 6c, with a similar working principle to that of CMP1.

#### 3.3.2. Bandgap Reference Circuit

The BGR is an essential component of LDO voltage regulators, which require a high power supply rejection ratio (PSRR) to provide constant voltage to implants. In our implantable application, a decoupling on-chip capacitor was integrated between the power supply and the output of the operational amplifier (opamp) to improve the PSRR of the BGR circuit [34]. This on-chip capacitor is highlighted in yellow in Figure 7.

The BGR circuit comprises a BGR central component, a folded-cascade amplifier, a startup circuit, and a control signal. The control signals, $V_{EN}$ and $V_{EP}$, function as a complementary pair of voltages to activate or deactivate the BGR circuit. The central part of the BGR circuit is composed of two branches, each consisting of a bipolar junction transistor (BJT) and a PMOS transistor. It is important to note that the impedance of $R_{3a}$ and $R_{3b}$ is designed to be the same, requiring careful layout considerations for proper matching. The inputs of the folded-cascade amplifier, denoted as $V_{+}$ and $V_{-}$, are designed to maintain almost identical voltages. The current $I_{b2}$ in the right branch of the BGR circuit can be obtained by subtracting the collector voltages of $Q_{1}$ and $Q_{2}$ and then dividing it by the impedance of $R_{4}$. The output voltage $V_{bg}$ is then determined by the expression $I_{b2}\cdot R_{3b}+V_{+}$. To deliver the output of the BGR circuit denoted as $V_{REF}$, a low-pass filter is typically employed to filter out any high-frequency noise or disturbances.

#### 3.3.3. Low-Dropout Regulator Circuit

The LDO circuit consists of an error amplifier, a power transistor, a buffer, and a load, as shown in Figure 8.

The main pole is located at the load, whereas the second pole corresponds to the output of the error amplifier, which connects to the gate of the power transistor. To handle a high current, a power transistor must be large in size. The large size of such a transistor, however, implies a large capacitor. As a result, it may create this second pole close to the main pole, which may give rise to instability issues. In order to address this issue and improve stability, a buffer is placed between the error amplifier and the power transistor. The buffer transistor ($M_{61}$) has a considerably smaller size than the power transistor ($M_{63}$), which extends the frequency range of the second pole and enhances stability.

The three LDO regulators were designed to deliver 1.2 V, 1.8 V, and 3.1 V. They operate independently, ensuring simultaneous output of their respective voltages with minimal or negligible interactions. For the error amplifier, one input is connected to the BGR’s output, $V_{REF}$, which provides a stable and reliable reference voltage. The other input monitors a fraction of the LDO regulator’s output voltage, obtained through an impedance divider, denoted as $V_{FB}$. By comparing $V_{REF}$ with the monitored $V_{FB}$, the error amplifier accurately controls the LDO regulator’s output voltage. This feedback mechanism ensures precise voltage regulation, allowing each LDO regulator to independently maintain its desired output voltage level.

### 3.4. Binary Phase-Shift-Keying-Based Data Communication

In BPSK, each bit of digital data is represented by a phase shift of the carrier signal. Specifically, a phase shift of 180° is utilized to represent the change in data between 1 and 0, or vice versa. The BPSK demodulator was designed based on the architecture reported in [35] and is illustrated in Figure 9.

The maximum data rate of the BPSK demodulator was set at 0.06 Mb/s, and it operates with a carrier frequency of 2 MHz. This demodulator consists of a self-biased PLL (SB-PLL) and digital demodulation circuits. The self-biased PLL is used to recover the carrier signal, which is used as the clock signal of the digital demodulation circuits. By setting the current of the current-control oscillator (CCO) proportional to the current of the charge pump (CPs) and using a source-degeneration-resistor-based voltage to current (V2I), a relevant constant loop bandwidth of our SB-PLL can be obtained. As a result, the SB-PLL can robustly operate over process, voltage, and temperature (PVT) variations, which enables a low supply voltage to save power consumption. The phase frequency detector (PFD) output signals (UP and DN) in the PLL are also used to detect if the date transition occurs by the trigger detector based on the pulse width comparison scheme. The data recovery block is used to obtain the demodulated data ($D_{DEM}$), and the resettable-divider-based clock recovery block is used to remove the phase shift from the data transition and obtain a clock with the same rate of $D_{DEM}$. Finally, the recovered data are retimed by the recovered clock. Thus, the synchronous clock and data signal are obtained for the whole implanted system.

### 3.5. Output Stage of Biphasic Current Stimulation

A biphasic current stimulating output stage (OS) was purposefully designed during the initial validation of the proposed EWHGNS system. The schematic diagram is illustrated in Figure 10.

Within this schematic, a 5 bit current controller was intricately designed to generate a 5 bit programmable current, directly correlated with the input code ($D_{0}$–$D_{4}$). The CCS is responsible for managing the activation and deactivation of this current generator. Subsequently, this current is duplicated using a current mirror circuit comprised of transistors $M_{5}$ through $M_{7}$. The control signals ($P_{1}$–$P_{4}$) are deployed to regulate the activation of transistors $M_{1}$ through $M_{4}$, facilitating the control of current flow in either direction through the hypoglossal nerve–electrode interface (HNEI) load. This HNEI is linked to two electrodes, indicated by the red and red dots.

This circuit can be simplified into the model depicted in the upper right block. When $P_{1}$ and $P_{4}$ are turned on, while $P_{2}$ and $P_{3}$ are turned off, current $I_{out}$ traverses from the red electrode (left) to the red electrode (right). Conversely, when $P_{1}$ and $P_{4}$ are turned off and $P_{2}$ and $P_{3}$ are turned on, current $I_{out}$ flows in the opposite direction. When $P_{1}$ and $P_{2}$ are activated and $P_{3}$ and $P_{4}$ are deactivated, the capacitor discharges into the power supply. Conversely, when $P_{1}$ and $P_{2}$ are deactivated and $P_{3}$ and $P_{4}$ are activated, the capacitor discharges into the ground. The power supply, denoted as $V_{OUT3}$, is derived from the output of the LDO regulator, set to provide a voltage of 3.1 V.

## 4. Results

The presented results can be divided into three parts: (1) regarding the external controller, the verification of the OSA detector was carried out using the CNN algorithm with ECG signals as raw data; (2) the results related to the WPT link were obtained through a combination of modeling, simulation, and experimental analysis; (3) for the implant part of the EWHGNS, the performance of the PMM, BPSK modulation scheme, and biphasic current stimulating output stage was evaluated based on post-layout simulations and experimental measurements.

### 4.1. OSA Detector Result

To validate and evaluate the performance of the proposed OSA detector, we conducted the experiment on the Apnea-ECG database [36], which is a public open-source dataset in PhysioNet. The dataset contains 70 individual ECG signals, recorded at a sampling rate of 100 Hz. Each ECG trace was labeled by an expert for apnea-related and normal annotations through minute-by-minute respiration and related signals. In the experiment, we randomly selected 28 records for the training dataset and 7 records for the test dataset. We segmented the recording signal into periods of 60 s and 120 s after the bandpass filter as the preprocessing function.

Table 4 shows the result analysis and comparison for the proposed method. In the experiments, the sensitivity (SEN), specificity (SPE), precision (PRE), accuracy (ACC), and F1-score were used to evaluate the network performance, which are expressed as below:(4)$$\begin{array}{cc} SEN & =\frac{TP}{TP+FN} \end{array}$$(5)$$\begin{array}{cc} SPE & =\frac{TN}{TN+FP} \end{array}$$(6)$$\begin{array}{cc} PRE & =\frac{TP}{TP+FP} \end{array}$$(7)$$\begin{array}{cc} ACC & =\frac{TP+TN}{TP+TN+FP+FN} \end{array}$$(8)$$\begin{array}{cc} F1 & =\frac{2PRE\times SEN}{PRE+SEN} \end{array}$$
where true positive (TP) represents the number of normal events correctly classified as normal, true negative (TN) represents the number of abnormal events correctly classified as abnormal, false positive (FP) represents the number of abnormal events incorrectly classified as normal, and false negative (FN) represents the number of normal events incorrectly classified as abnormal. During the validation process, given the sample length of 60 s, the model obtained an accuracy of 86.74%, a sensitivity of 93.52%, a specificity of 84.23%, and an F1-score of 85.95%. It achieved comparative results over many traditional methods as shown in the table. Furthermore, increasing the segment length of the recorded signal led to an improvement in the classification results. The high sensitivity in the validation dataset indicated the system’s capacity to accurately issue alarms upon detecting abnormal breathing states, which is of great importance in timely and appropriate clinical intervention.

### 4.2. Wireless Power Transfer Link Results

In our study, we utilized COMSOL to construct a 2D axisymmetric model consisting of two concentric coils, as is shown in Figure 11. The primary coil, referred to as Tx, was implemented using the Coil Setting in COMSOL. It was modeled as a homogenized multiturn conductor, with the number of turns set to nine. Similarly, the receiving coil, denoted as Rx, had a similar configuration to Tx with the number of turns set to four. To assess the SAR, we performed simulations by applying a current of 1 A to the primary coil, Tx. The SAR values were then evaluated within the model. Figure 11 shows that the SAR value for a tissue with a mass of 0.5 kg was 0.001 W/kg using the proposed coil parameters. In our simulation, the SAR values were indeed expected to be higher in the vicinity of the excited coil, where the electromagnetic fields are more concentrated.

To enable WPT, customized inductive coils were used. These coils were created with the aid of a high-precision automatic winding machine, as illustrated in Figure 12. The electrical parameters of the coils were obtained by converting the scattering parameters measured using a Keysight ENA Network Analyzer E5071C. The mutual and self-inductance values were derived from the S-parameters. The transmitting coil had a diameter of 35 mm, while the receiving coil had a diameter of 9.2 mm. In the case of a 5 mm pork thickness, which simulates the skin, the measurement showed that s21 was −4.99 dB and its corresponding PTE was 31.8%.

Table 5 shows a comparison of our inductive WPT system for biomedical implants with other studies reported in the literature. In the previous study [40], a receiver coil with a ferrite core, having a diameter of 22 mm, was utilized, resulting in a high power transfer efficiency (PTE) of 76.3%. However, the use of a ferrite core could result in a high value of the SAR, which was not provided in [40]. This high SAR level can potentially have adverse effects on human tissue. In contrast, a millimeter-sized receiver coil was proposed in [41], but the achieved PTE was limited to 1.4%, with a coupling value of only 3.5 $\times \;10^{-3}$. Another study [42] achieved a PTE of 18% for the WPT link.

In the current study, the PTE achieved was lower compared to the previous study [40], with a value of 38.2%. However, in this study, the SAR was effectively managed and maintained at a limited level of only 0.001 W/kg, without the use of a ferrite core. Under the SAR limit of 2 W/kg for 10 g of tissue, the maximum power delivery to the load in this paper was 20 mW, much higher than the value of 2.2 mW in [40]. The utilization of a smaller receiver coil diameter of 9 mm in this study posed a challenge in achieving a high PTE.

### 4.3. Implant Results

#### 4.3.1. Power Management Module

To verify the operation analysis of the active-doubler-based rectifier introduced in Figure 6, a schematic simulation was performed, and the transient results are displayed in Figure 13. When the voltage $V_{VD}$ is lower than the IGND, the comparator CMP1 detects this condition and switches its output, $V_{CN}$, to a high-voltage level. This change in voltage activates the transistor $M_{1}$, causing current to flow from the IGND to $V_{IN}$ through $M_{1}$, resulting in the charging of capacitor $C_{3}$. Similarly, when the voltage $V_{VD}$ exceeds $V_{REC}$, another comparator, CMP2, senses this condition and switches its output, $V_{CP}$, to a low-voltage level. This low-voltage level triggers the main power transistor $M_{3}$, allowing current to flow to the load. In the simulation, the input voltage amplitude was set to 1.6 V, and the operating frequency was 2 MHz. The output of the rectifier measured 2.78 V with a load impedance of 1 kΩ.

Furthermore, a post-layout simulation was conducted to analyze the transient behavior of the PMM during startup. The simulation results are illustrated in Figure 14. Initially, the power transistor $M_{1}$ acted as a diode when the input voltage $V_{IN}$ was higher than $V_{REC}$. As the load capacitor charged over time, the rectifier switched into active operation, functioning as a doubler-rectifier. In this simulation, the input voltage amplitude was set to 2.6 V, and the output of the rectifier reached 3.3 V. Additionally, the PMM incorporated three LDOs providing output voltages of 1.26, 1.83, and 3.14 V, respectively. Each LDO was connected to a load with a resistance of 1 kΩ.

The PMM components of the NIHGHS were fabricated using the 40 nm process available at Taiwan Semiconductor Manufacturing Co. LTD (TSMC). The total area, without the PADs, is 543 μm × 117 μm. A photomicrograph of the PMM is shown in Figure 15. The measurement setup for the PMM is illustrated in Figure 16. It consisted of an arbitrary function generator and a mixed-signal oscilloscope (Keysight MSOS604A).

The voltage waveform resulting from the measurement of the rectifier is presented in Figure 17. It operated at a frequency of 2 MHz, with an AC input amplitude of 1.18 V. Under these conditions, the rectifier generated a DC output of 2 V when connected to a load impedance of 1 kΩ. However, due to the coupling of input noise to the $V_{VD}$ node and the charge and discharge of the load capacitor, an output ripple voltage was observed in the rectifier. As shown in Figure 17, the rectifier exhibited a peak-to-peak ripple of 0.15 V.

In Figure 18, the relationship between the measured power conversion efficiency (PCE) and voltage conversion ratio (VCR) as a function of the load impedance is plotted. The PCE represents the ratio of the power received by the receiving coil to the power drawn by the load, expressed as follows:(9)$$\eta_{{}_{\mathrm{REC}}}=\frac{P_{OUT}}{P_{AC}}=\left|\frac{\frac{V_{OUT}^{2}}{R_{L}}}{\frac{1}{Td}\int_{t_{0}}^{t_{0}+Td}V_{AC}\left(t\right)\cdot I_{AC}\left(t\right)dt}\right|$$
where $V_{AC}$ is the amplitude of the input AC voltage, $I_{AC}$ is the amplitude of the input AC current, and $V_{OUT}$ is the amplitude of the output DC voltage. The VCR is defined as the ratio of $V_{REC}$ to $V_{IN}$. At an input amplitude of 1.18 V and a load impedance of 1 kΩ, the maximum value of VCE was 1.69. However, the PCE was only 77.5% under these conditions, which is suboptimal. The PCE reached its optimal value of 79% at a load impedance of 1.2 kΩ. As the load impedance increased beyond this point, the PCE showed a declining trend, while the VCE did not vary significantly. For this measurement, we maintained $V_{REC}$ constant at 2 V and the frequency at 2 MHz, while tuning the load impedance and input voltage.

Figure 19 illustrates the measured PCE of the active-doubler-based rectifier at various operating frequencies ranging from 0.5 to 5 MHz. The maximum amplitude of the input voltage, denoted as $V_{IN}$, was set to 1.18 V, and the load impedance was 1 kΩ. It was observed that, as the operating frequency increased, a noticeable degradation in the PCE occurred, particularly beyond 2 MHz. This degradation can be attributed to the increasing coupling effect between the $V_{VD}$ and $V_{CN}$ nodes, resulting from the presence of a parasitic capacitor in the circuit.

The stronger coupling effect at higher frequencies disrupted the normal switching operation of transistor $M_{1}$, which was controlled by the comparator CMP1. As a consequence, the efficiency of power conversion was compromised. Similarly, the presence of a parasitic capacitor between the $V_{VD}$ and $V_{CP}$ nodes introduced a stronger coupling effect, further deteriorating the performance of the rectifier at higher operating frequencies. The abnormal switch behavior of the power transistor $M_{3}$ controlled by another comparator aggravated the decrease in the PCE. Managing coupling effects and minimizing the impact of parasitic capacitances are critical challenges in the design of an active rectifier, particularly when aiming to improve the PCE at a higher operating frequency.

The relationship between the input voltage supply and the output voltage of the LDO is illustrated in Figure 20, which was designed to deliver a voltage of 3.1 V.

Based on the measurement results, it was observed that, when the voltage supply varied between 2.5 V and 3.2 V, while keeping the load impedance at a constant value of 1 kΩ, the output exhibited a linear increase. At a supply voltage of 3.2 V, the output voltage reached 3.09 V. Furthermore, as the supply voltage continued to increase beyond 3.24 V, the LDO maintained a stable output voltage of 3.1 V, which indicated a voltage drop of 0.13 V. This demonstrated that, within a certain range of input voltages, the LDO can regulate the output voltage at 3.1 V effectively.

We conducted measurements of the output of LDO3 at various load currents, using a power supply with a constant voltage of 3.3 V. When the load current was 0.1 mA, the LDO3 output was measured to be 3.17 V, which dropped to 3.1 V as the load current increased to 3 mA. The output ripple was measured to be 0.06 V when the load current varied from 0.1 mA to 3 mA, as is shown in Figure 21. The maximum PCE of LDO3 was achieved when it delivered a load current of 3 mA, at which point, the LDO3 output voltage was 3.1 V. It was calculated as the ratio of the output power to the input power. This maximum PCE was estimated to be 93.9% (3.1 V/3.3 V) × 100%, under the assumption that the current of other branches in LDO3 was negligible compared to the load current of 3 mA. The overall PCE of this proposed PMM was calculated to be 72.9% under the load impedance of 1 kΩ. The calculation is provided as:(10)$$\eta_{PMM}=\eta_{REC}\times \eta_{LDO}$$

Table 6 presents a comprehensive comparison between previous power-management systems for implantable devices and the proposed PMM. The PMM offers significant improvements over the system described in [40], which had a limited output voltage of 1.8 V. In contrast, the PMM provides a wider range of voltage options, including 1.2, 1.8, and 3.1 V, increasing the flexibility and adaptability of the device. One notable achievement of the PMM is its significantly reduced chip area, measuring just 0.06 mm${}^{2}$ (excluding PADs), in contrast to the dimensions reported in [40] (0.12 mm${}^{2}$) and [43] (0.3 mm${}^{2}$). This significant reduction in size was achieved through the utilization of smaller power transistors within the active-doubler-based rectifier and LDO regulators. This downsizing did result in a slight decrease in the PCE of the PMM. In terms of the PCE, the active voltage doubler presented in [44] achieved the highest efficiency of 92.2%. However, it did not integrate LDOs nor offer the integrated chip area.

#### 4.3.2. Binary Phase-Shift Keying Building Block

Firstly, this proposed PLL-based BPSK demodulator was verified through simulation. The results of the post-layout simulation with a data rate of 62.5 kb/s are presented in Figure 22. As shown in this figure, the recovered synchronous data and 50% duty-cycle clock signal are denoted as $D_{OUT}$ and $CK_{OUT}$, respectively, which were separately recovered by the BPSK demodulator. The modulation signal is represented by $D_{IN}$, while $B_{IN}$ denotes the carrier signal after modulation. To improve waveform clarity, a short period of time for $B_{IN}$ was trimmed.

To further validate the functionality of the proposed BPSK demodulator, a proof-of-concept prototype featuring discrete components was designed. The phase frequency detector (PFD) was constructed using a D-type flip-flop SN74AUP1G79, while the self-biased charge pump was assembled using components including a current generator (ALD1116), a current mirror (ALD1117), and two switches (ADG609BN). The voltage-controlled oscillator was created with a MAX2471, and the digital circuitry was implemented through a digital XOR gate, an inverter, and multiple D-type flip-flops.

Figure 23 shows the input data of the modulated BPSK digital signal and demodulated data. It is worth noting that, due to the electrical characteristics of the discrete components, the power supply utilized in this test bench was set at 3.3 V. The results indicated a successful demodulation of the data using this architectural configuration.

The BPSK demodulator presented in this work had a low power consumption of 42 μW at a 0.7 V supply voltage. It is worth noting that this supply voltage was derived by tapping a fraction of the output voltage from an LDO regulator that provides a stable 1.2 V. As shown in Table 7, this work achieved lower power consumption than the recently reported BPSK demodulators. Moreover, this demodulator exhibited excellent energy efficiency compared to PLL-based BPSK demodulators for high-Q coil telemetry. Although the PLL-less BPSK demodulators in [45] exhibited better energy efficiency due to their high data rates, they were not suitable for high-Q telemetry owing to their open-loop structure.

#### 4.3.3. Output Stage of Biphasic Current Stimulation

The simulation results are presented in Figure 24. “Dis_Gnd” signifies that the capacitor discharges into the ground, while “Dis_Vdd” indicates discharging into the power supply. “Neg” denotes that the current flows from right to left in Figure 10, whereas “Pos” signifies a current flow in the opposite direction. Clearly, the simulation results showed that the voltage between the two electrodes accurately reflects the expected current flow direction.

A proof-of-concept prototype, constructed using discrete components, was devised to validate the proposed architecture. The 5 bit current controller was realized through the use of multiple NMOS current mirrors (ALD1116), while the PMOS current mirror was implemented with the ALD1117. Furthermore, four switches were fashioned using ADG609BN. The waveforms obtained from the measurements are depicted in Figure 25.

To evaluate the initial functionality of the stimulator, a load comprising a 1 kΩ resistor in series with a 400 nF capacitor was employed under stimulation currents of 1.2 mA. The results conclusively demonstrated that this structure fulfills the biphasic function requirements essential for the proposed EWHGNS system.

## 5. Conclusions

We developed a miniaturized ECG- and WPT-based hypoglossal nerve stimulation microsystem as an innovative solution for addressing patients with OSA. This device integrates several essential components, including an external OSA detector based on ECG, customized winding coils, a power management module, a BPSK demodulator, and a biphasic current stimulator. The ECG-based OSA detector demonstrated a high accuracy of 88.68% in monitoring episodes of apnea. The design of the inductive link coils took into consideration factors such as the SAR limitations, PTE, and coil size to maximize overall performance. The WPT link achieved a PTE of 31.8% with a 5 mm distance between the transmitter and receiver coils. The receiver coil had a diameter of 9 mm. The PMM consisted of an active-doubler-based rectifier, a bandgap reference, and three LDOs. It delivered three DC voltages required for different components and achieved a power conversion efficiency (PCE) of 74% while occupying an area of only 0.25 mm${}^{2}$. The BPSK demodulator, based on PLL technology, exhibited a low power consumption of 42 μW and supported a data rate of 0.0625 Mb/s. Currently, a new version of the system is being undertaken to complete the whole device and corresponding packaging.

## Figures and Tables

**Figure 1 sensors-23-08882-f001:**
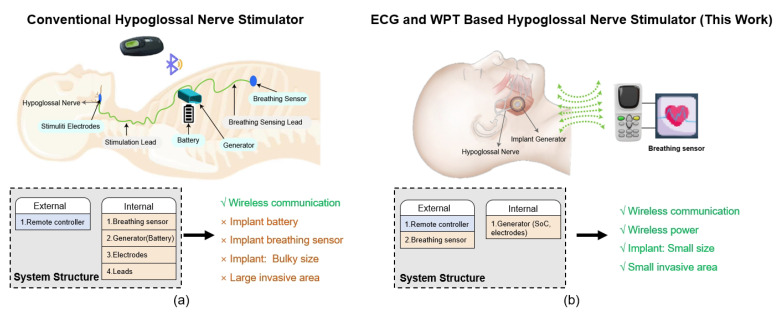
Hypoglossal nerve stimulator. (**a**) Conventional device [6]. (**b**) Proposed ECG- and WPT-based system.

**Figure 2 sensors-23-08882-f002:**
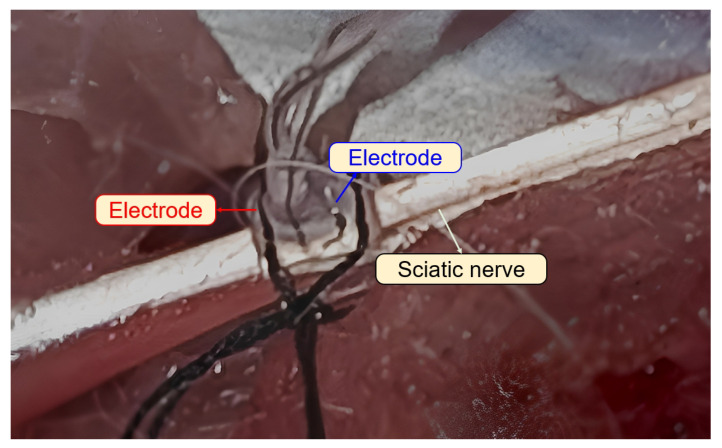
Experimental setup for measuring the sciatic nerve–electrode interface in animal testing using a rabbit model.

**Figure 3 sensors-23-08882-f003:**
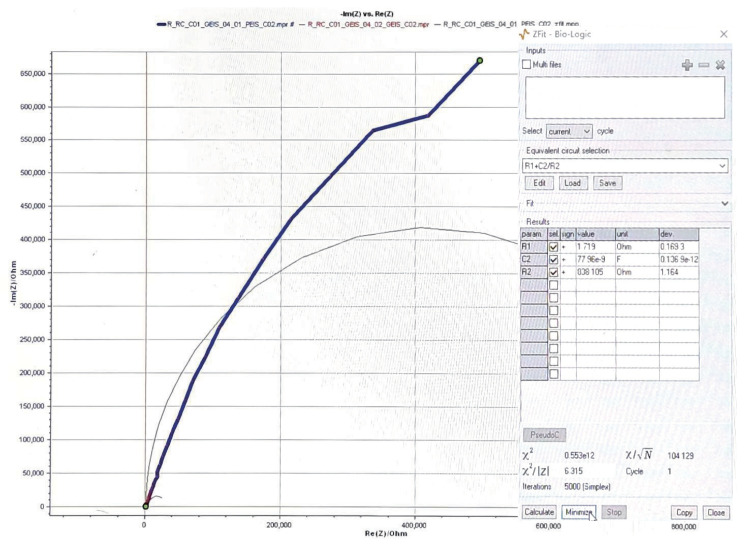
Measured impedance waveforms between two electrodes and the equivalent circuit parameters.

**Figure 4 sensors-23-08882-f004:**
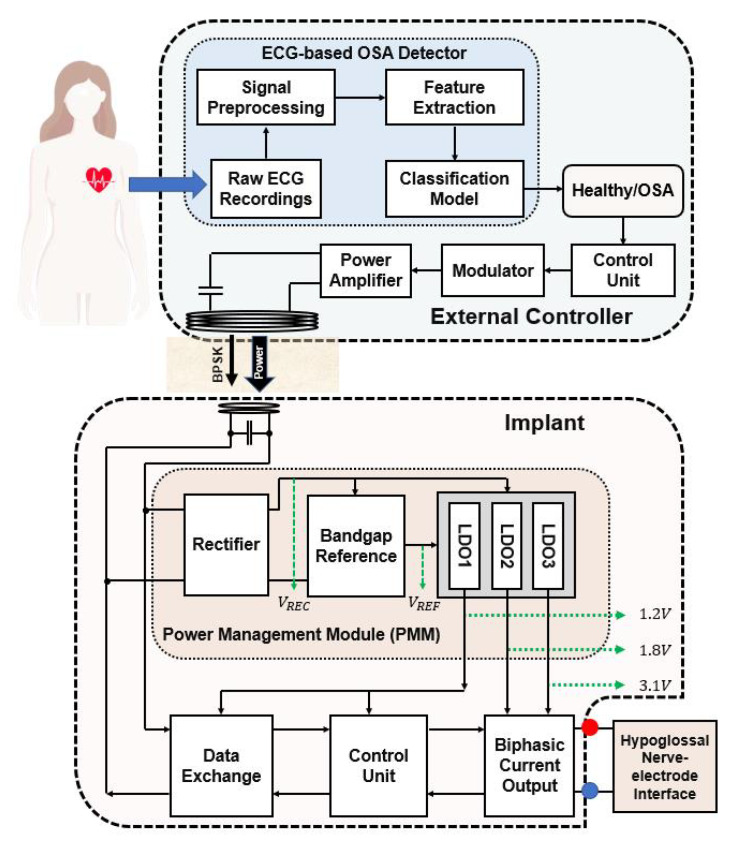
Block diagram of the proposed ECG- and WPT-based implantable hypoglossal stimulation system.

**Figure 5 sensors-23-08882-f005:**
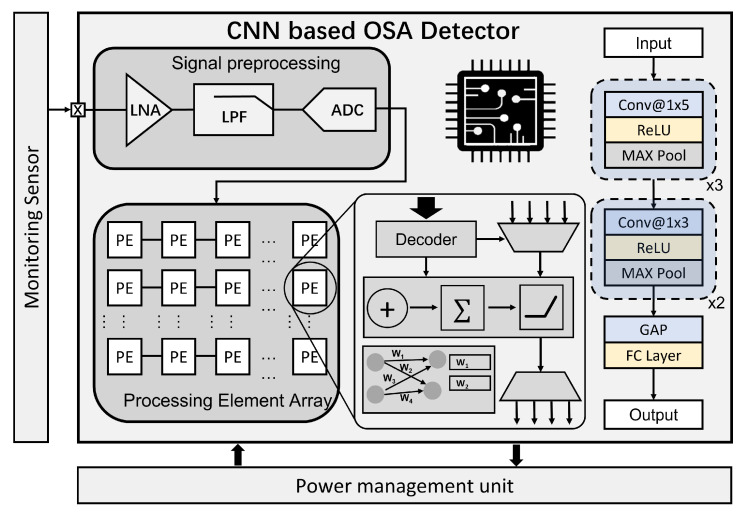
Diagram of the proposed OSA detector. Conv: convolutional layer. GAP: global average pooling layer. FC: fully connected layer.

**Figure 6 sensors-23-08882-f006:**
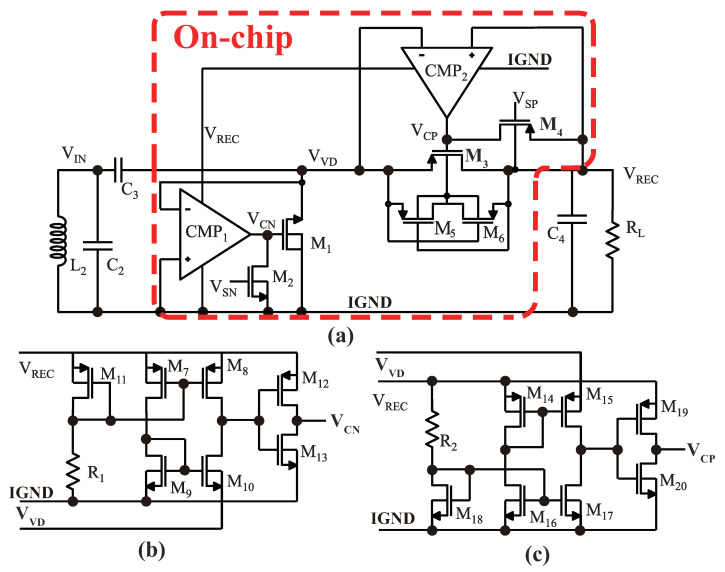
Schematics of the active doubler–based rectifier: (**a**) rectifier connects with internal off–chip coil and capacitors; (**b**) schematic of CMP1; (**c**) schematic of CMP2.

**Figure 7 sensors-23-08882-f007:**
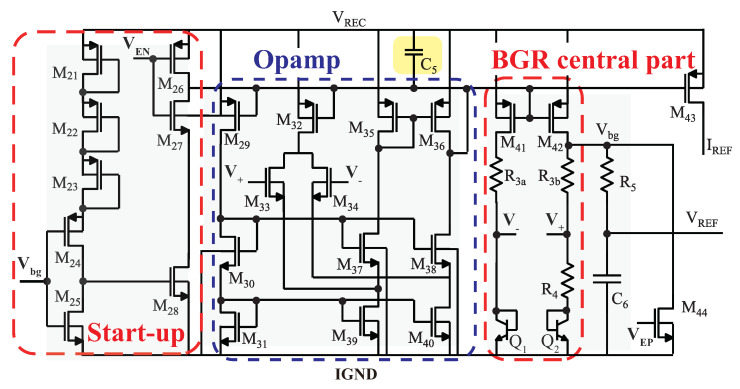
Schematic of the BGR circuit.

**Figure 8 sensors-23-08882-f008:**
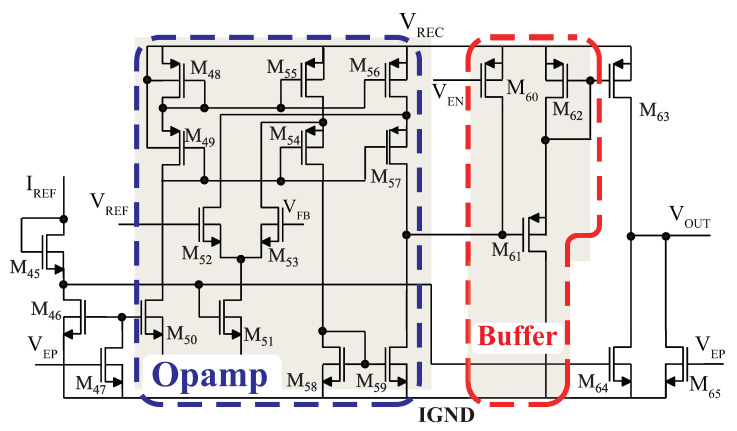
Schematic of the LDO circuit.

**Figure 9 sensors-23-08882-f009:**
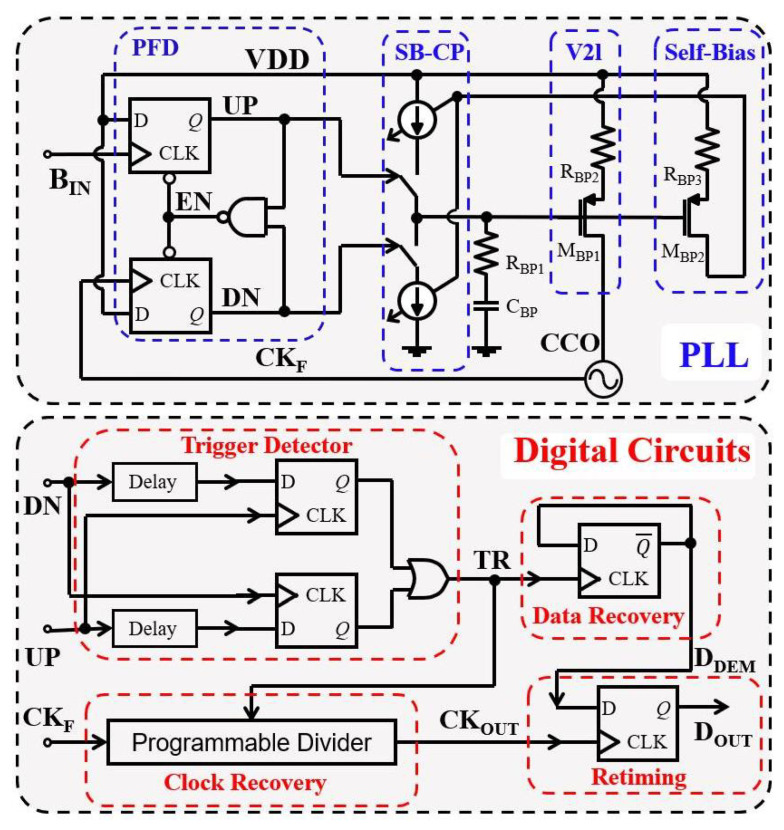
Schematic of the PLL-based BPSK demodulator.

**Figure 10 sensors-23-08882-f010:**
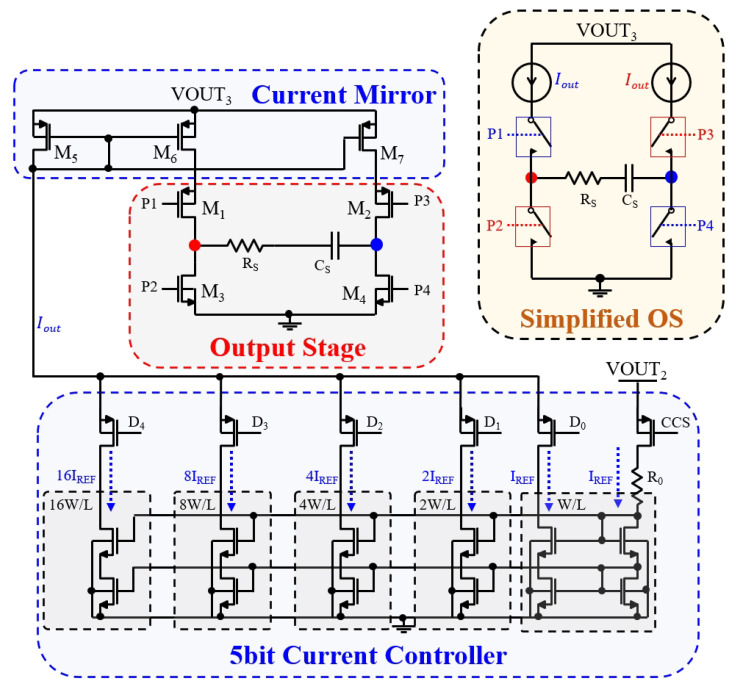
Schematic of the bipolar biphasic current stimulator.

**Figure 11 sensors-23-08882-f011:**
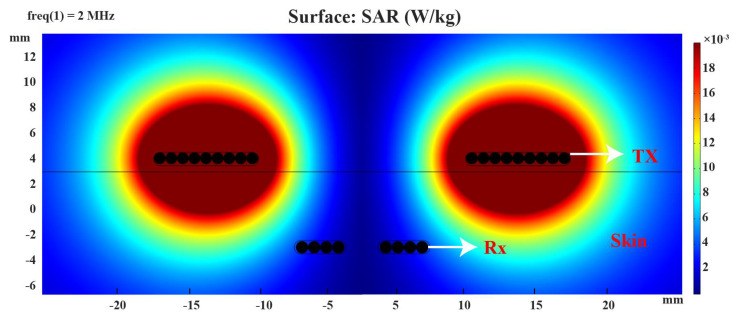
Simulated specific absorption rate (SAR) distribution from lateral view.

**Figure 12 sensors-23-08882-f012:**
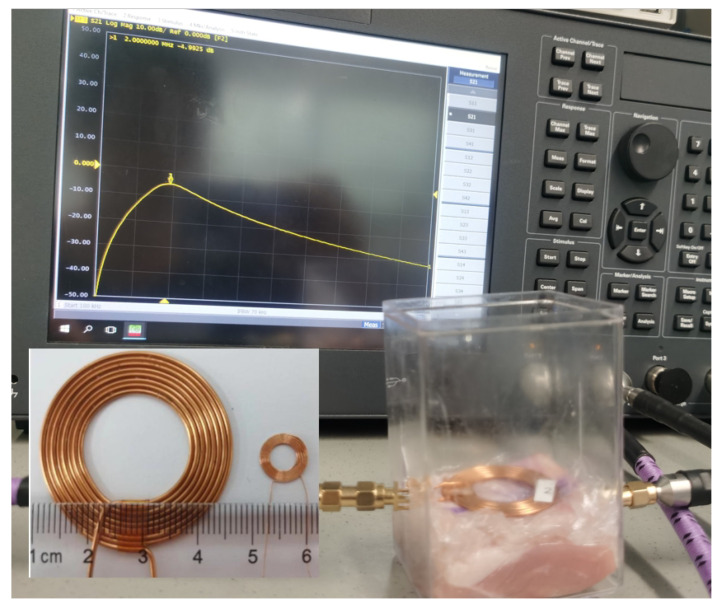
Measurement result of inductive link.

**Figure 13 sensors-23-08882-f013:**
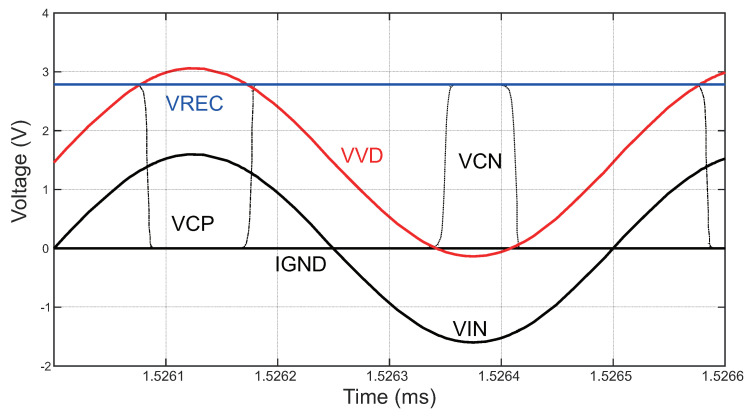
Simulated waveforms of the rectifier circuit.

**Figure 14 sensors-23-08882-f014:**
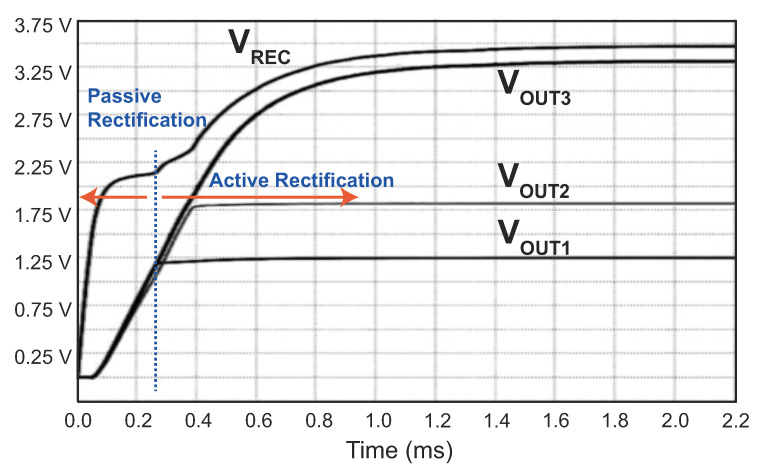
Post-layout simulation results of the power management module.

**Figure 15 sensors-23-08882-f015:**
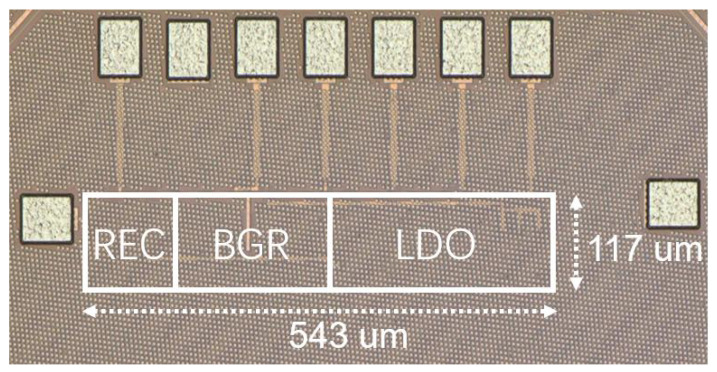
Chip photograph of the power management module.

**Figure 16 sensors-23-08882-f016:**
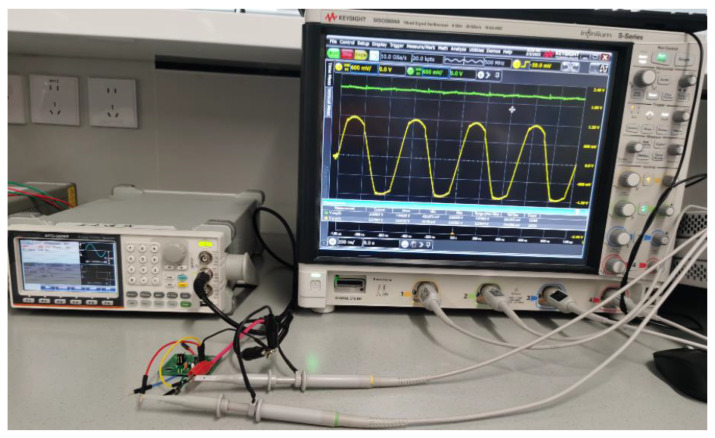
Measured setup of the power management module.

**Figure 17 sensors-23-08882-f017:**
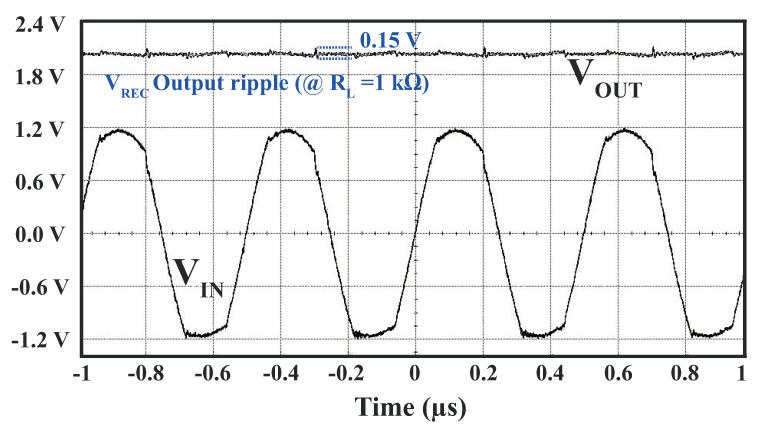
Measured waveforms of the proposed active-doubler-based rectifier.

**Figure 18 sensors-23-08882-f018:**
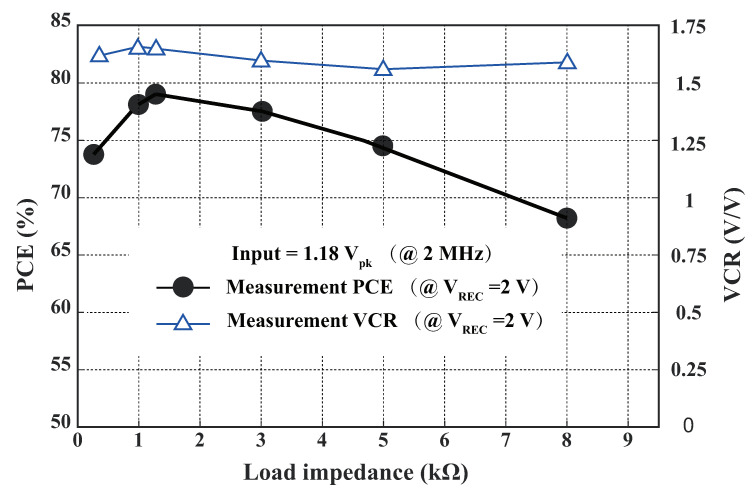
Measured PCE and VCR of the proposed active-doubler-based rectifier with different load impedances.

**Figure 19 sensors-23-08882-f019:**
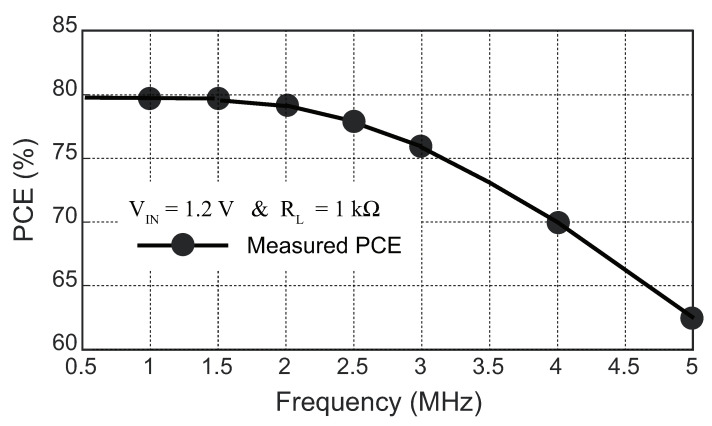
Measured PCE of the proposed active doubler-based rectifier with different frequencies.

**Figure 20 sensors-23-08882-f020:**
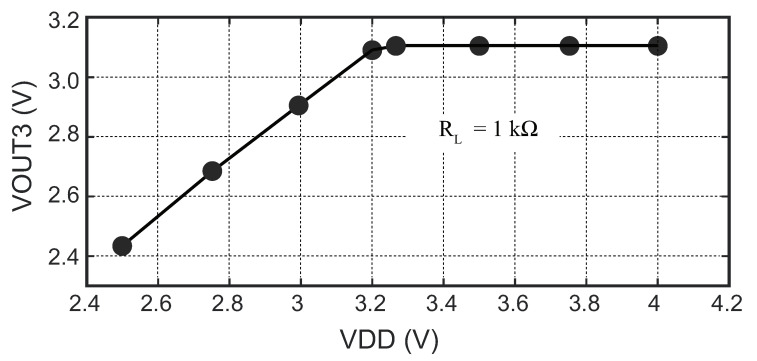
Measured output voltage of LDO3 with different supply values.

**Figure 21 sensors-23-08882-f021:**
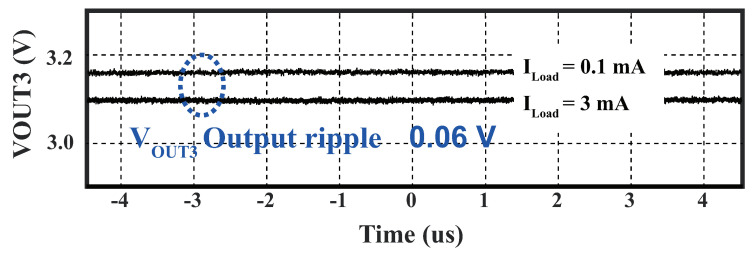
Measured waveforms of LDO3 with two load values.

**Figure 22 sensors-23-08882-f022:**
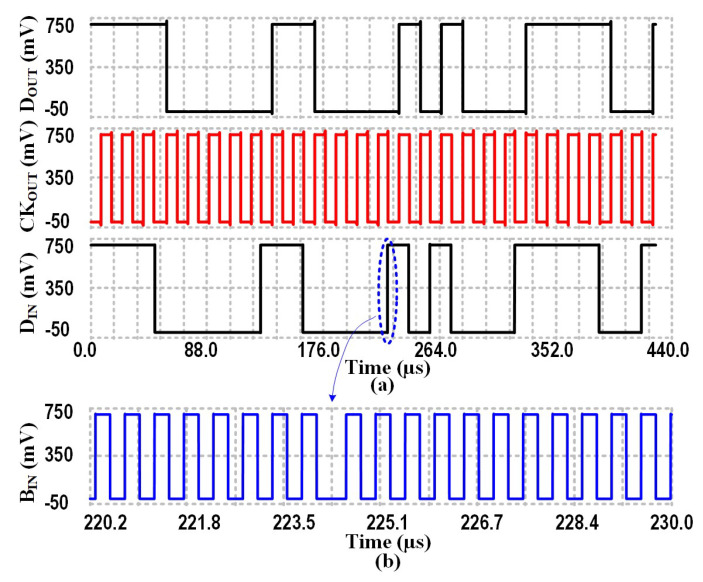
Post-layout simulation results of the proposed BPSK circuit, and (**b**) represents the amplification of the dashed line in (**a**).

**Figure 23 sensors-23-08882-f023:**
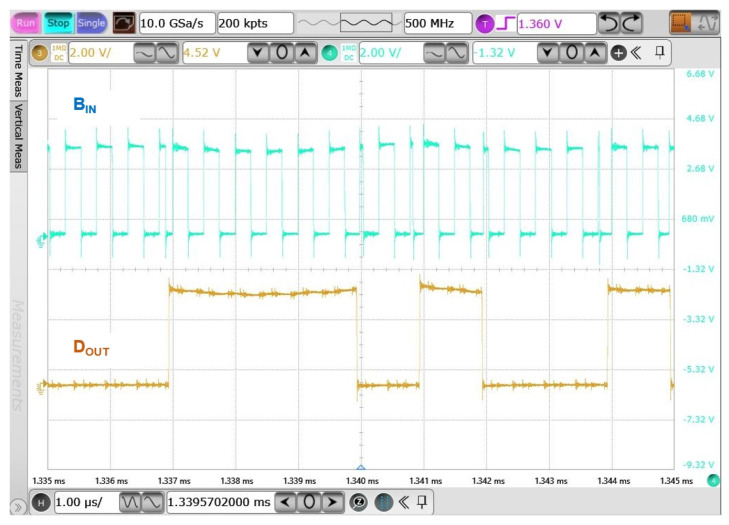
Measurement results of BPSK demodulator.

**Figure 24 sensors-23-08882-f024:**
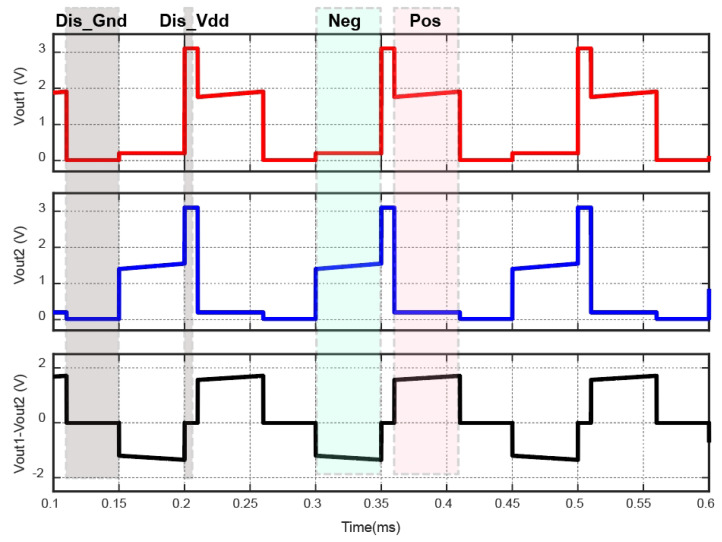
Output voltages of the proposed biphasic current stimulator.

**Figure 25 sensors-23-08882-f025:**
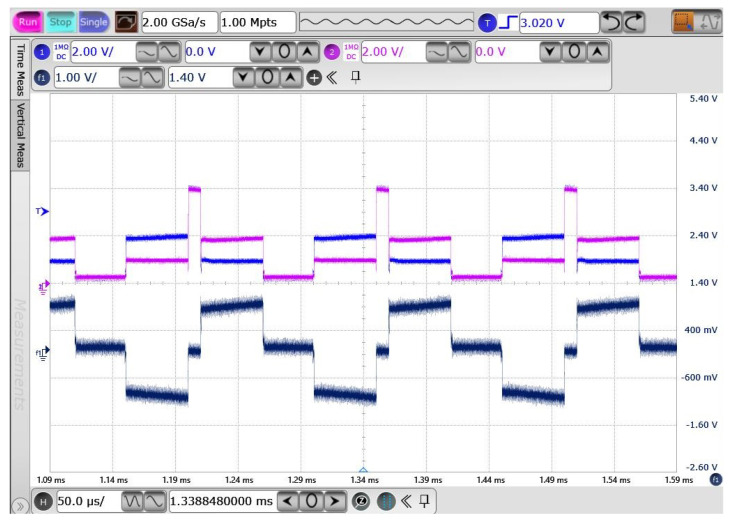
Measurement results of the proposed biphasic current stimulator.

**Table 1 sensors-23-08882-t001:** Comparison of different breathing-monitoring methods.

Methods	Advantages	Limitations	Ref.
PSG	Accurate	Complex instrument; laboratory testing; costly and time-consuming	[11,12]
ECG	Accurate	Need algorithm to analyze data	[13,14]
Sound Sensing	Non-contact	Low accuracy; susceptible to noise; in R&D stage	[12,13,15]
Remote-Based	Non-contact	Sensitive to interference from the subject’s movements and activities; in R&D stage	[12,15]

**Table 2 sensors-23-08882-t002:** Main parameters of the proposed EWHGNS.

Parameters	Value/Technique	
**OSA monitor**	ECG	
**Wireless power transfer**	Inductive link	
	RX (size, dimension)	≈10 mm
	D (Skin separating TX and RX)	5 mm
	Carrier frequency	2 MHz
**Data exchange**	Modulation type	BPSK
	Information stream	10 bit
**Power management**	Input (Volt)	AC (1–3.3)
	Output(Volt)	DC (1.2; 1.8; and 3.1)
**Stimulation stage**	Power supply (Volt)	3.1
	Stimulus type	Biphasic current
	Frequency range	10–40 Hz
	Current density range	0–2.4 mA
	Digital/analog converter	5 bit
	Number of channels	1
	Nerve–electrode interface	1 kΩ, 400 nF

**Table 3 sensors-23-08882-t003:** The parameters of an inductive coupling link.

Name	Expression	Description
*k*	$M/\sqrt{L_{1}L_{2}}$	Coupling coefficient of two coils
$Q_{1}$	$\omega L_{1}/R_{1}$	Quality factor of the primary coil
$Q_{2}$	$\omega L_{2}/R_{2}$	Quality factor of the secondary coil
$Q_{L}$	$R_{L}/\omega L_{2}$	Quality factor of the load
ω	2π*f*	Angular frequency

**Table 4 sensors-23-08882-t004:** Performance analysis for period apnea event.

Method	Segmentation	SEN (%)	SPE (%)	PRE (%)	ACC (%)	F1 (%)
KNN [37]	60 s	82.00	82.00	81.45	82.21	81.72
SVM [38]	60 s	81.24	69.84	72.90	75.54	76.84
RNN [39]	60 s	97.00	87.00	84.70	85.40	71.79
**Filter + CNN**	**60 s**	**88.64**	**85.72**	**83.42**	**86.74**	**85.95**
	**120 s**	**93.52**	**83.73**	**84.23**	**88.68**	**88.63**

**Table 5 sensors-23-08882-t005:** Comparison of WPT link for implantable device (IMD).

Description	[40]	[41]	[42]	This Work (Mea.)
Tx: $Q_{1}$	245	289	139	**88**
Rx: $Q_{2}$	161	31	22	**10**
Coupling: k	0.1	3.5 $\times \;10^{-3}$	N/A	**0.09**
PTE (air): $\eta \left(\%\right)$	76.3	N/A	N/A	**39.8**
PTE (skin): $\eta \left(\%\right)$	N/A	N/A	18	**31.8**
Dia. of Rx (mm)	22	1	2.5	**9.2**
Frequency (MHz)	13.56	20	60	**2**
Distance (mm)	N/A	10	5	**5**
Max PDL (mW)	N/A	2.2	N/A	**20**

**Table 6 sensors-23-08882-t006:** Comparison with other power management blocks in implantable device.

Publications	[40]	[43]	[44]	This Work
CMOS process (μm)	0.18	0.5	0.35	**0.04**
Structure	Doubler LDO	Adaptive Active	Doubler	Doubler LDO
Frequency (MHz)	13.56	2	6.78	**2**
$V_{IN,peak}$ (V)	1.192	5	1–1.7	**1.4–2.6**
$V_{REC}$ (V)	2	2.5–4.6	3.3	**3.3**
$V_{OUT,LDO}$ (V)	1.8	N/A	N/A	**1.2, 1.8, 3.1**
$V_{dropout}$ (V)	0.38	N/A	N/A	**0.13**
PCE (%)	85	72–87	92.2	**77.5**
$R_{L}$ (kΩ)	0.1	$I_{L}$ = 2.8 mA	0.5	**1**
Chip Area (mm${}^{2}$) (Without PAD)	0.12	0.3	N/A	**0.064**

PCE: power conversion efficiency; $R_{L}$: load impedance.

**Table 7 sensors-23-08882-t007:** Comparison with prior BPSK demodulator.

Publications	[46]	[45]	[29]	This Work
CMOS Technology (μm)	0.18	0.18	0.5	**0.04**
Carrier Freq. (MHz)	13.56	2	13.56	**2**
Type	PLL-Based	PLL-Less	PLL-Based	**PLL-Based**
High-Q Telemetry	Yes	No	Yes	**Yes**
Power Consumption (μW)	217	82	5148	**42**
Date Rate (Mb/s)	0.211	1	0.02	**0.06**
Supply Voltage (V)	2	1.8	3.3	**0.7**
Energy Efficiency (pJ/bit)	1028	82	257,400	**672**

## Data Availability

Data sharing not applicable.

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
