# Peer review of "Minimally Invasive Hypoglossal Nerve Stimulator Enabled by ECG Sensor and WPT to Manage Obstructive Sleep Apnea"

_sensors, 2023, doi:10.3390/s23218882_

Round 1
Reviewer 1 Report
Comments and Suggestions for Authors
I congratulate the authors for the work done. It has been very interesting for me to read this article.
Author Response
We would like to express our sincere gratitude for your valuable comments. We have improved our paper in the attachment.

Reviewer 2 Report
Comments and Suggestions for Authors
The paper presents the deveolpment of a new device for treating obstructive sleep apnea. The proposed hypoglossal nerve stimulator consists of an electrocardiogram sensor that detect the obstructive sleep aprnea through a CNN classifier, a wireless power transfer, a power management module, a binary phase-shift keying based data communication and a biphasic current stimulating output stage. The authors present both the implementation requirements and the design of each individual block. A simulation shows the results obtained with the proposed system.
The reviewer greatly appreciated the paper as it is an interesting topic and the work done by the authors is of quality. However, some modifications are recommended to improve the presentation of the work.
The major change the authors should make is to move some parts found in the results (section 4) to the methods part (section 3). An example is lines 436-443, which should be moved in section 3.1. More generally, all initial rows of each results subsection should be moved to the corresponding subsection of Section 3.
Additionally, I reproduce below some suggestions:
- an example/reference is recommended for the continuous positive airway pressure at line 32;
- I would suggest considering to cite recent references regarding the use of photopletismography for ECG monitoring since they can provide a valuable contribution in the present study:
Almarshad, Malak Abdullah, et al. "Adoption of Transformer Neural Network to Improve the Diagnostic Performance of Oximetry for Obstructive Sleep Apnea." Sensors 23.18 (2023): 7924.
Vicente-Samper, José María, et al. "An ML-Based Approach to Reconstruct Heart Rate from PPG in Presence of Motion Artifacts." Biosensors 13.7 (2023): 718.
- it would be better to specify that the reference figure is Figure 1(a) at line 40, while Figure 1(b) at line 65;
- it is reccommended to enter the accuracy values in Table 1 instead of the + signs. In addition, a column should be included to specify the method used to obtain the specified accuracy;
- a reference is recommended at line 125;
- it would be better to remove the parentheses in lines 171-172 because the impedance names have already been defined previously in lines 156-157;
- it is recommended to specify who the various quantities are in equation 1;
- you should replace "2 classifications" in line 301 with "2 classes";
- lines 314 to 316 are not very clear: it would be better to write them differently, perhaps in list form;
- the validation dataset discussed in section 4.1 is actually a test dataset;
- it is recommended to specify who the various quantities are in equation 4.
Author Response
We would like to express our sincere gratitude to your valuable comments. Every suggestion has been considered, and we have made significant revisions to improve our paper. Please see the attachment.

Reviewer 3 Report
Comments and Suggestions for Authors
The manuscript is well-written and was designed properly. I recommend the acceptance of the manuscript
Comments on the Quality of English Languageminor errors
Author Response
We would like to express our sincere gratitude for your valuable comments. Enclosed, please find our enhanced paper as an attachment.
Q1: Minor errors in the quality of the English language.
Authors A1: We have updated this paper in the following Lines.
Lines 1-2: “Hypoglossal nerve stimulator (HGNS) is an invasive device that is used to treat obstructive sleep apnea (OSA) through electrical stimulation”.
Lines 2-3: “The conventional implantable HGNS device consists of a stimuli generator”.
Line 19: “The functionality of the proposed ECG and WPT-based HGNS was validated”.
Line 34: “a bulky CPAP device”.
Besides, we added spaces between number and unit, such as:
Line 9: “With a skin thickness of 5 mm and a receiving coil diameter of 9 mm”.
Lines 14-15: “it offers three-voltage options (1.2 V, 1.8 V, and 3.1 V)”.
Line 97 “0.06 mm2”.
Line 98: “(1.2 V, 1.8 V, and 3.1 V)”.
Line 100: “42 μW”.

Reviewer 4 Report
Comments and Suggestions for Authors
In this manuscript, the authors presented a ECG-based hypoglossal nerve stimulator for treating obstructive sleep apnea. The article is well-written, but some minor revisions are suggested.
1. In Table 1, accuracy of each methods was described as “+”. Could the authors provide specific percentage values for accuracy for each method? This would offer more straightforward information for readers to evaluate each method. Additionally, some of the references, such as Ref. 11 and 12, are from five years ago. Are there any updated references published?
2. Figure 2 is unclear. Please consider replacing it with one that has better resolution.
3. To assess the impedance of the HNEI value, the authors used an adult rabbit. I wonder if the value is the same for humans?
4. In line 445, the authors wrote, "the accuracy (ACC), sensitivity (SEN)…, precision (PRE)." Could you explain the differences between accuracy, sensitivity, and precision? Also, there is no precision data shown in Table 3.
5. Since the device has not been tested inside an animal, could the authors discuss some possible issues that might arise during an animal test experiment and how to address these issues?
Author Response

(The authors gave the same response as above.)
